# Small Ultrasound-Based Corrosion Sensor for Intraday Corrosion Rate Estimation

**DOI:** 10.3390/s22218451

**Published:** 2022-11-03

**Authors:** Upeksha Chathurani Thibbotuwa, Ainhoa Cortés, Andoni Irizar

**Affiliations:** 1CEIT-Basque Research and Technology Alliance (BRTA), Manuel Lardizabal 15, 20018 Donostia-San Sebastián, Spain; 2Department of Electronics and Communications, Universidad de Navarra, Tecnun, Manuel Lardizabal 13, 20018 Donostia-San Sebastián, Spain

**Keywords:** corrosion monitoring, corrosion rate, FPGA, offshore wind turbines, ultrasound, thickness loss

## Abstract

The conventional way of studying corrosion in marine environments is by installing corrosion coupons. Instead, this paper presents an experimental field study using an unattended corrosion sensor developed on the basis of ultrasound (US) technology to assess the thickness loss caused by general atmospheric corrosion on land close to the sea (coastal region). The system described here uses FPGA, low-power microcontroller, analog front-end devices in the sensor node, and a Beaglebone black wireless board for posting data to a server. The overall system is small, operates at low power, and was deployed at Gran Canaria to detect the thickness loss of an S355 steel sample and consequently estimate the corrosion rate. This experiment aims to demonstrate the system’s viability in marine environments and its potential to monitor corrosion in offshore wind turbines. In a day, the system takes four sets of measurements in 6 hour intervals, and each set consists of 5 consecutive measurements. Over the course of 5 months, the proposed experiment allowed for us to continuously monitor the corrosion rate in an equivalent corrosion process to an average thickness loss rate of 0.134 mm/year.

## 1. Introduction

As the offshore wind-energy industry is prepared to increase its capacity in the following years [1] it is also facing important challenges. The need for wind farms farther away from land (10–50) km and at increasing depths (>200 m) [2] has led to new wind tower designs and deployment strategies [3] to reduce capital expenditure (CAPEX). However, in the medium and long term, wind-farm operators face increasing costs due to the operation of the wind farms in extremely harsh environments. Maintenance tasks in offshore wind farms need to be thoroughly prepared in advance because access to the wind towers is expensive in terms of both trained staff and trip costs [4].

Therefore, structural health monitoring (SHM) has become one of the major disciplines for offshore wind turbines (OWTs) with the necessity of improving operation and maintenance strategies [5]. SHM provides a strategy for the damage detection of structures in the mean of changes in material or geometric properties that affect current or future performance [6]. Hence, the continuous structural health monitoring of wind turbines would be beneficial in improving structural reliability and optimizing maintenance tasks at minimal associated costs [7]. Corrosion is one of the main roots for degradation of offshore structure materials which could eventually lead to damage and weaken the structure [8]. The deterioration of the materials and their properties due to corrosion is a fact that reduces the useful service life of the material and leads to fail the structures, equipment and other objects from their intended functioning. Studies show that the estimated cost of corrosion is about 3–4%, of each nation’s gross domestic product (GDP). It was estimated that between 15% and 35% of the cost of corrosion could be saved through improved corrosion management [9].

### Corrosion Monitoring in Offshore Wind Turbines

Corrosion in the tower and other equipment within the wind tower is one of the main consequences of an offshore environment [10]. In particular, corrosion in the tower and its foundations is critical because it could potentially lead to a structural failure of catastrophic consequences. The form of corrosion on offshore steel wind turbine towers varies for each tower zone depending on the access to the levels of oxygen, humidity, and water, including water water temperature, salinity, and depth [11].

Corrosion coupons are the most conventional technique for calculating the corrosion rate in an offshore environment. Corrosion rates for construction steel below sea level are 0.2 mm/year on average; in tidal and splash zones, the rate may fluctuate from around 0.4 to 1.2 mm/year [12]. Reported corrosion rates of steel from different marine locations are available in [13]. Moreover, a 0.83 mm/year of corrosion was experimentally estimated in the North Sea, which is quite a higher value compared to other reported values [14].

Hence, the early detection of the possible risks of structural failure allows for avoiding severe damage to the structure and saving on associated costs. For that, maintenance operators are very keen on proper and efficient solutions that can continuously monitor corrosion unattended while using wireless connectivity to obtain real-time data, with the possibility to move around the tower to inspect new areas as required.

Corrosion monitoring covers a broad range of techniques that involve measuring and converting a measured parameter into corrosion loss or rate [15,16,17]. On the basis of the sensing principle, different types of corrosion detection sensors are available [18]. The change in the material could be sensed via the loss of weight, the alteration of physical, chemical, electrical, magnetic, or mechanical properties, and the loss of component integrity (e.g., cracking). Nondestructive testing (NDT)-based corrosion detection allows for inspecting the material without disturbing the material properties. Hence, NDT techniques are more suitable for corrosion monitoring solutions with real-time data acquisition systems and remote data logging.

Ultrasound techniques for corrosion monitoring are based on the continuous reduction in the thickness of a material immersed in a corrosive environment. As such, they are normally classified as a direct, nondestructive, and nonintrusive technique. Nonintrusiveness means that the sensor itself does not influence the corrosion mechanism taking place. Furthermore, the ultrasound sensor can be located on the “clean” side of the material being corroded. There are many commercial ultrasound devices that provide valuable information about various forms of degradation (cracks, deformations, thinning, corrosion, etc.). However, in general, they are classified as inspection devices rather than monitoring devices. Inspection means measurements carried out by trained staff over a predetermined time frame in accordance with maintenance schedules. Instead, corrosion monitoring devices perform very frequent measurements with the aim of detecting small fluctuations, in this case related to corrosion phenomena. Thus, the data coming from the corrosion monitoring devices can be used as input for a general assessment of corrosion. The precision obtained with ultrasound signals allows for very frequent thickness measurements that can be used to track corrosion.

In this paper, we present a miniaturized corrosion monitoring solution based on ultrasound technology for offshore platforms. The ultrasound probe was glued onto an S355 bare steel sample of 5 mm in thickness. The developed system was deployed at a corrosive environment in Gran Canaria with the aim of analyzing the thickness loss evolution and estimating the corrosion rate in a real and harsh scenario.

This paper is organized as follows. The theory of the ultrasound-based sensor is presented in Section 2, including the theory of ultrasound with ToF estimation in Section 2.1, the description of the corrosion sensor in Section 2.2, the discussion of the ToF calculation algorithm with the temperature compensation in Section 2.3, and how to calculate the relative thickness loss is discussed in Section 2.4. Next, the introduction to the experiment of measuring thickness loss due to corrosion in real conditions is provided in Section 3, followed by the experiment’s results and discussion in Section 4.

## 2. Theory

### 2.1. Ultrasound Theory

In the ultrasound technique, the corrosion loss rate is determined on the basis of wall thickness loss. During the ultrasound test, a short duration of ultrasound energy is introduced into the test object at periodic intervals of time. The sound energy propagates through the material in the form of mechanical vibrations. This propagation attenuates and weakens the sound wave, with the travel distance mainly as a result of scattering, absorption, and reflection [19]. When a sound wave strikes a medium interface, part of the energy is transmitted into the next medium across the boundary, while some is reflected into the first medium as a result of different acoustic impedances in different media or materials. The amount of reflected or transmitted energy gives information about the size of the reflector.

On that basis, there are two approaches that can be used to estimate thickness during ultrasound testing: through-transmission and pulse-echo techniques. In the through-transmission technique, the transmitted part of the signal through the test specimen is monitored, whereas in the pulse-echo technique, the reflected signal at different boundaries is considered [20]. Hence, in through transmission, two transducers are necessary for transmission and reception, whereas in the pulse-echo technique, a single transducer is enough for both tasks.

Generally, ultrasound nondestructive-testing-based measurements involve wave propagation (transit) time across a given distance and a degree of attenuation that takes place at that time [21]. Measurement modes used to find the transit time of a sound wave in the process of thickness calculation can be classified on the basis of the choice of echoes [22]:Mode 1: excitation signal—first back-wall echo.Mode 2: near surface—first back-wall echo.Mode 3: two successive back-wall echoes.

The probes used to generate and receive ultrasonic energy are known as ultrasonic transducers. Typically, these transducers are produced from piezoelectric ceramic or composite material [23]. The performance efficiency of a piezoelectric transducer depends on the proper matching of electrical and acoustic impedance. Electrical impedance matching needs the design of an electrical matching circuit between the driving circuit and the transducer; the acoustic impedance between the piezoelectric generators/detectors and the propagating media likewise needs to be matched. A mismatch could lead to a loss in signal transmission and a low signal-to-noise ratio (SNR) [24]. The electrical impedance can be matched in different ways, such as using an impedance matching electrical network, and matching the characteristics and properties of the cable connected to the transducer [25]. A significant acoustic impedance mismatch results in most of the signal’s energy being reflected. Hence, a couplant material (often a liquid, such as a thin film of oil, glycerin, or water) is typically required in nondestructive material testing to reduce the acoustic impedance mismatch between air and a test specimen.

In our solution, we used the pulse-echo technique with a single direct-contact transducer, and Mode 3 was selected to estimate the thickness. Moreover, relative measurements were performed to observe the thickness loss in a certain location where the sensor was placed, which we discuss further in Section 2.4.

The capacity of the ultrasound technique was presented in [26], measuring the wall thickness loss of a mild steel sample under accelerated laboratory conditions. Our approach presents a novel solution, applying the ultrasound technique to a smart continuous monitoring system deployed on a marine environment over the course of several months.

#### Estimation of Time of Flight

Using ultrasonic waves, corrosion condition is evaluated on the basis of the wall thickness loss caused by corrosion. Generally, the two common ways to determine the geometric distance measurement on the basis of an ultrasound response are phase shift and time of flight (ToF) [27,28].

ToF is the time taken by ultrasound signal to arrive from transmitter to receiver corresponding to the thickness *d* of the test specimen; when a single transducer is used, this relation can be stated as in Equation (Equation 1).
(1)d=c×ToF/2
where *c* is the speed of sound in the material. This value depends on the physical properties of the material/medium (elastic moduli and the density of the material) [29]. The most common technique employed in many ultrasonic applications to estimate a signal’s transit time through a test material is the ToF technique, which was also applied to our approach.

Applying a threshold with an analog comparator at a certain amplitude level is the most fundamental and straightforward method of calculating the ToF value. A more accurate and reliable way is to apply appropriate digital signal processing techniques. In that case, the ultrasound response is processed on the basis of its echo parameters. One of the most common methods is to consider the received signal as a delayed and attenuated (amplitude-scaled) version of a reference signal with noncorrelated additive noise [30]. The objective is to estimate a ToF measurement by matching the received signal with a reference signal. For that, the cross-correlation function is widely used to mathematically quantify the highest similarity between the reference and delayed version of signals/echoes in the time domain. An estimation of the ToF value is the time at which the cross-correlation result reaches its maximum.

However, the precision of a ToF measurement depends on the quality of the received signal. ToF value estimation can become more complex in the presence of noise, interference, scattering, and attenuation of the signals [31]. Thus, it may require a preprocessing step of filtering before the cross-correlation process. Even afterwards, ambiguities detecting the actual ToF value in the cross-correlated result are possible; in such situations, the precision of the ToF estimation could be improved with additional computational steps [31,32,33,34].

Our solution uses a single transducer based on the pulse-echo method. As the same transducer is employed for transmission and reception, it is important to consider the dead zone of the sensor, which is the interval of the transducer ringing due to the initial pulse and not yet completely ready to receive/detect reflected echoes [35]. Figure 1 shows a raw signal obtained for a bare steel sample of 5 mm thickness. There is an interval of approximately 440 samples (3.5 μs) produced by the piezoelectric sensor’s dead zone, in which the output of the sensor presented transient behavior due to the characteristics of the signal’s generation circuitry and the electrical adaptation inside the piezoelectric sensor. Since the first back-wall echo falls into the dead zone of the piezoelectric sensor, the next two consecutive back-wall echoes were used to calculate the ToF value (using Mode 3 as mentioned in Section 2.1).

### 2.2. Description of the Corrosion Sensor

Our miniaturized and low-cost monitoring solution (see the electronics in Figure 2) is expected to operate unattended under the harsh conditions of offshore platforms for a long period, estimating the thickness loss due to corrosion. A small size (110 × 60 × 15 mm), wireless connectivity, and low power (5.4 μW in standby mode, 850 mW average power for measurement events with a duration of 200 μs) are some important features to facilitate deployment and to provide higher autonomy. On top of that, the low weight of the system (around 100 g) is important to obtain a feasible solution integrated into a mobile platform with the aim of covering large structures by using the same sensor node.

The ultrasound probe/transducer used in the proposed solution is V111 from Olympus [36] with a peak frequency of 8.44 MHz and 15 mm of diameter of contact. The sensor was chosen after analyzing the performance specifications prioritizing its lower waveform duration (higher bandwidth) and sound pressure power.

#### Architecture of the Sensor Node

Typically, an ultrasound testing system consists of a pulser, a transmitter/receiver transducer, an amplifier, and data collection and display devices/platform. The hardware architecture of our corrosion sensor comprised a low-power Cortex-M4-based microcontroller interfacing with an Intel MAX 10 FPGA by using a serial peripheral interface (SPI) link and analog front end (AFE) devices as shown in Figure 3. More details related to the microcontroller and the wireless connectivity of the system are provided in [37].

When the microcontroller requests a ToF measurement, the field-programmable gate array (FPGA) takes control of the devices that form the data acquisition circuit [36]. Thus, the FPGA is responsible for generating enabling signals for the ultrasound pulser and the trigger to start the data acquisition. The ultrasound pulser circuit used in this work can generate high-voltage, high-frequency, unipolar, or bipolar pulses. In the this paper’s experiment, the excitation pulse is a bipolar square wave with 50% duty cycle. The pulse frequency in MHz was set according to Equation (Equation 2), where 125 MHz is the sampling frequency and FREQC was set as an even positive integer to maintain a duty cycle of 50%.
(2)f=125/FREQC

Once a measurement is performed, the FPGA processes the acquired data and provides the calculated ToF estimation to the microcontroller. This signal processing is controlled by the microcontroller through the design parameters explained in Table 1 and shown in Figure 4 over the filtered ultrasound response. The LOC and PEAK values shown in Figure 4 are calculated in the FPGA and are used internally to extract the different echoes. They are also accessible via SPI as read-only registers.

The ultrasound response processed by the FPGA was previously amplified by a low-noise variable gain amplifier (VGA) that was composed of two cascaded amplifiers, with each one providing a maximal amplification of up 38 dB. Both amplifiers were controlled with analog voltage V_ctrl_ from 0.2 to 2.3 V. The relation between the control voltage V_ctrl_ and the total amplification G is given by Equation (Equation 3).
(3)$$G(\mathrm{dB})=44\times V{}_{\mathrm{ctrl}}-23.6$$

The system is capable of generating the V_ctrl_ for the amplifier using the microcontroller and allowing for further degrees of flexibility of its operation. The corresponding signal level of the ultrasound response (root mean square (RMS) value of the first detected echo) is measured by the FPGA and sent to the microcontroller via SPI with the aim of implementing an automatic gain controller. The signal processing algorithm for ultrasound response analysis takes place in the FPGA and is explained next in Section 2.3.

Furthermore, the sensor node reads from a 16-bit analog-to-digital converter (ADC) (LT2451) [38] through an interintegrated circuit (I2C) to link temperature values to the corrosion measurements using the NTC10K3A1I thermistor [39] as the temperature sensor to acquire the temperature of the steel sample. The variation of ToF with temperature is discussed in Section 2.4.

### 2.3. Algorithm Description of ToF Calculation

The algorithm extracts two consecutive back-wall echoes from the received ultrasound response with an echo windowing process that separates the echoes for the ToF estimation. The thickness at a certain time and position is determined by using the ToF technique as given in Equation (Equation 1). The size of the S355 bare steel sample to test the algorithm was 75×150×5 mm. Therefore, considering the initial thickness of the sample to be 5mm, the expected delay between echoes was $2\times 5/5.9\times 10^{6}=$ 1.69 μs (the speed of sound in steel is approximately $5.9\times 10^{6}$ mm/s).

A block diagram of the digital processing stages performed inside the FPGA to calculate the ToF value between two consecutive echoes is given in Figure 5. The signal from the piezoelectric sensor was first bandpass (BP)-filtered; then, the envelop of that signal was obtained. Both signals, the outputs of the bandpass filter, and the envelop filter are shown in Figure 4.

The two extracted consecutive echoes (see Figure 4), Echos 2 and 3 from the bandpass signal, were cross-correlated to detect the point of maximal matching to determine the ToF value. In the cross-correlation approach, the obtained time delay at the maximal cross-correlation peak was used for estimating this ToF value. Generally, time delays in cross-correlation results are not integral multiples of the sampling period. Therefore, the largest cross-correlation peak position could be between the indices of the time-delay vector. Thus, to improve the precision of the ToF value, a cross-correlation function (ccf) can be interpolated between the samples [40]. In our solution, parabolic interpolation was used to estimate the location of the ccf peak with the aim of reducing the computational complexity. The final cross-correlation output from the second and third consecutive echoes is shown in Figure 6a. To obtain the final ccf peak, curve fitting was performed for the maximal cross-correlation peak, and its two nearest neighbors were as shown in Figure 6b. The peak offset (δToF) was estimated using parabolic interpolation given in Equation (Equation 4), where r[0] is the largest peak of the cross-correlation result, and r[−1] and r[+1] are its two nearest neighbors.
(4)$$\delta \mathrm{ToF}=\frac{0.5\times (r[-1]-r[+1])}{r[-1]+r[+1]-2\cdot r[0]}$$

Accordingly, the final cross-correlation peak after parabolic interpolation was obtained at δ ToF. In the example shown in Figure 6b, the maximal correlation peak was obtained at correlation index −10. Therefore, the estimated delay between the two echoes was −10+δToF.

The methodological design presented in [35] was followed to validate this digital processing. The first step was to use our own testbed on the basis of Red Pitaya [41] to acquire ultrasound signals. Then, a model was developed in MATLAB to process the ultrasound responses. Afterwards, the digital processing algorithm was implemented in Red Pitaya and compared against the MATLAB model. Lastly, this signal processing was implemented into the FPGA and verified using the MATLAB model.

### 2.4. Calculation of Relative Thickness Loss Using ToF

The measurement of ToF in ultrasound signals propagating through steel is directly related to variation in the speed of sound with temperature and, to a lesser extent, to thermal expansion. Therefore, we needed to measure the temperature of the steel to compensate for this temperature effect in the measurement of ToF. Let us assume that, in the absence of corrosion, we were measuring the ToF of a bare steel sample of thickness $L_{s0}$ at a room temperature $T_{0}$ and the speed of sound $v_{s0}$ also at $T_{0}$. For the relatively small temperature ranges that we considered, the propagation speed of the ultrasonic signal in steel at a given temperature *T* follows this expression:(5)$$\begin{array}{c} v_{s}\left(\Delta T\right)=v_{s0}\cdot \left(1+\xi_{1}\Delta T+\xi_{2}\Delta T^{2}+\cdots \right), \end{array}$$
where $\xi_{i}$ are constants of the material (°C${}^{-i}$), $\Delta T=T-T_{0}$ is the temperature difference with respect to $T_{0}$. For the range of working temperatures, 15 °C<T<25 °C, we can safely assume that a first-order approximation is more than enough, and $\xi =\xi_{1}$. At a given temperature *T*, the measured ToF is:(6)$$\begin{array}{c} \mathrm{ToF}=\frac{2L_{s}}{v_{s}}=\frac{2L_{s0}\left(1+c_{x}\Delta T\right)}{v_{s0}\left(1+\xi \Delta T\right)}={\mathrm{ToF}}_{0}\frac{1+c_{x}\Delta T}{1+\xi \Delta T}, \end{array}$$
where $c_{x}$ is the thermal expansion coefficient of steel ($c_{x}≊12\times 10^{-6}\;$°C${}^{-1}$). The value of ξ for steel is small and negative, so the speed of sound decreases as temperature increases.
(7)$$\begin{array}{cc} \mathrm{ToF} & \approx {\mathrm{ToF}}_{0}\left(1+c_{x}\Delta T\right)\left(1-\xi \Delta T\right)={\mathrm{ToF}}_{0}\left[1+\left(c_{x}-\xi \right)\Delta T\right]\\ \mathrm{ToF}/{\mathrm{ToF}}_{0} & \approx 1+(c_{x}-\xi )\Delta T\\ \frac{\Delta \mathrm{ToF}}{{\mathrm{ToF}}_{0}} & \approx (c_{x}-\xi )\Delta T \end{array}$$

The value of $\left(c_{x}-\xi \right)$ can be experimentally determined by running an experiment that measures the ToF in a bare steel sample in a climatic chamber (see Section 2.4.1).

In the initial state, the bare sample presented no corrosion, and the first ToF measure was taken at temperature *T${}_{0}$*. We call that value ${\mathrm{ToF}}_{0}$, which corresponds to an initial thickness $L_{s0}$:(8)$$\begin{array}{c} {\mathrm{ToF}}_{0}=\frac{2L_{s0}}{v_{s0}} \end{array}$$

If we now consider the situation of a corroded sample at temperature *T*, the thickness of that sample is related to ToF as follows:(9)$$\begin{array}{cc} L_{n} & =\frac{\mathrm{ToF}\times v_{s}}{2}=\frac{\mathrm{ToF}}{2}\times v_{s0}\left(1+\xi \Delta T\right)=\frac{\mathrm{ToF}}{2}\times \frac{2L_{s0}}{{\mathrm{ToF}}_{0}}\left(1+\xi \Delta T\right)\\ \; & =\frac{\mathrm{ToF}}{{\mathrm{ToF}}_{0}}\times L_{s0}\left(1+\xi \Delta T\right) \end{array}$$

However, $L_{n}$ is the thickness after corrosion at temperature *T*. We must reference all measurements to the same temperature *T${}_{0}$*. Thickness $L_{n}$ is related to the thickness after corrosion at temperature *T${}_{0}$*, $L_{n0}$, as follows: (10)$$L_{n}=L_{n0}\left(1+c_{x}\Delta T\right)$$

Therefore, the loss of thickness due to corrosion at temperature *T${}_{0}$* substituting Equation (Equation 9) into the above equation is:(11)$$\begin{array}{cc} \frac{\Delta L_{c}}{L_{s0}} & =\frac{\mathrm{ToF}}{{\mathrm{ToF}}_{0}}\times \left(1+\left(\xi -c_{x}\right)\Delta T\right)-1\\ \; & =\frac{\Delta \mathrm{ToF}}{{\mathrm{ToF}}_{0}}+\frac{\mathrm{ToF}(\xi -c_{x})\Delta T}{{\mathrm{ToF}}_{0}} \end{array}$$

#### 2.4.1. Temperature Experiment

To experimentally determine the value of ξ, we conducted an experiment that consisted of placing the sample in a climatic chamber and linearly varying the temperature from 20 to 40 °C. The process has to be as slow as possible to avoid transient behavior in the time-of-flight and temperature measurements. In this way, the process is almost stationary. Because corrosion is an even slower process, from the above equation, we can write that, for this experiment,
(12)$$\begin{array}{c} \frac{\Delta L_{c}}{L_{s0}}=0 \end{array}$$

Substituting the value of $\Delta L_{c}/L_{s0}$ into Equation (Equation 11), we obtain:(13)$$\frac{\Delta \mathrm{ToF}}{\mathrm{ToF}}=-\left(\xi -c_{x}\right)\cdot \Delta T=r\cdot \Delta T$$

Therefore, if we plot Equation (Equation 13) in the (Δ*T*,ΔToF/ToF) axis using the results of the experiment and perform a linear fitting, the calculated slope is $r=-(\xi -c_{x})$. With the obtained data (see Figure 7), we estimated the value of $\xi =-1.054\cdot 10^{-4}\;$°C${}^{-1}$.

To understand the needed amount of correction given the known values for ξ and $c_{x}$, this results in a correction in thickness of around —0.6 µm/°C for a 5 mm thick steel sample.

## 3. Experiment: Evolution of Thickness Loss Due to Corrosion in Real Conditions

With the aim of analyzing the evolution of thickness loss due to corrosion in a real environment, an experiment was carried out deploying our ultrasound sensor node in Gran Canaria, near the coast (see Figure 8).

The experimental setup was arranged by permanently adhering the ultrasound probe onto a certain spot on the (75×150×5) mm bare steel sample (as shown in Figure 9) using bicomponent epoxy adhesive Structalit 1028 R from Panacol [42]. After fixing the sensor to the sample, a bicomponent layer of epoxy resin PX900D [43] was applied over the sensor-attached side of the sample as a corrosion protection layer. As is shown in Figure 10b, the sensor was attached from the inner (clean) side of the sample, and the other side was fully exposed to the marine environment. Since the sensor was fixed to one particular location of the sample, this experiment allowed for monitoring the thickness loss due to corrosion, avoiding larger uncertainties due to thickness variations on different spots of the same sample. The thickness variations underneath the whole sensor (contact area of 15 mm) were even smaller than those when measuring different spots. The solution is averaging a shot of quick measurements. As was said in [35], thickness at different locations of the same sample varied even for the unexposed bare steel samples. These small thickness variations (at the µm level) were due to common existing imperfections during the sample production process. Therefore, in order to achieve more precise ToF measurements, the positioning of the sensor on the same location is very important.

To easily collect data from this experiment, the ultrasound sensor node was connected to a Beaglebone black wireless board (see Figure 10b). The Beaglebone was used to receive the ultrasound signals and to post the data to a data server in json format. Both the sensor node and Beaglebone are battery-powered (rechargeable, 3.6 V/7000 mAh), and the amplifier gain was set to 17.7 dB in this experiment.

Although our sensor node is capable of autonomously operating without a Beaglebone, in this case, the microcontroller from the sensor node communicated with the Beaglebone via the Universal Asynchronous Receiver and Transmitter (UART) protocol. The Beaglebone was basically in charge of sending the updated TOFREF value (see Table 1), estimated from the previous measurement event, to the sensor node at the beginning of an ultrasound measurement event, and then receiving and posting the data to the cloud after each measurement event. Thus, in a day, the system performed four measurement events (every 6 h). Each event consisted of five successive ToF measurements.

The corrosion monitoring setup was deployed at the PLOCAN premises in Gran Canaria (near the coast, see Figure 10) from the end of February to the beginning of August to receive thickness loss data, and to estimate the intraday corrosion rate thanks to the fast continuous measurements and high-precision estimates. Figure 11 shows how the bare steel sample looked like after five months of exposure in this marine environment.

The system gathered data for approximately 5 months, performing four measurement shots/events per day, and each shot was composed of five successive measures. Table 2 presents the main features of the experimental setup explained throughout this paper.

## 4. Results of the Experiment and Discussion

The corrosion rate is the speed of deterioration of a material due to corrosion. One of the most important aspects of the proposed corrosion monitoring solution is being able to perform very frequent measurements in order to calculate the corrosion rate. The proposed experiment was carried out, analyzing the thickness loss evolution due to uniform corrosion in a real marine environment, and estimating the corrosion rate in real time (for practical purposes, taking into account that corrosion is an extremely slow process).

After computing the average of the five successive measurements taken every 6 h, we processed a linear fitting of the last 28 measurements, which represent 1 week of data. Figure 12 shows the thickness estimations in mm obtained during the experiment.

During the first month, we chose the location and prepared the system setup. Once the experiment had begun, we could see that the measured temperature was higher than what was expected due to the setup being exposed to direct sunlight. We corrected that during May by shadowing the setup. This means that we measured higher temperatures from March to May (particularly during sunlight periods), but taking into account that the error for a 5 mm thick steel sample is —0.6 µm/°C, the measurements could still be considered to be acceptable. On the other hand, we lost data at the beginning of July because of the accumulation of dirt and dust on the cable of the ultrasound probe. Once the cable had been cleaned, the system resumed working as usual.

The thickness estimations were corrected on the basis of the reference temperature; in this case, we chose them to be the initial temperature that we had measured at the beginning of the experiment (21.3 °C). The correction to temperature variations was applied using the second term of Equation (Equation 11) and considering that $\left(\xi -c_{x}\right)$ was $-1.174\times 10^{-4}$ °C${}^{-1}$.

As commented in Section 2.3, we followed the methodological design presented in [35] to validate the system processing. Figure 13 compares the ToF estimations performed by the deployed system and the MATLAB model using the raw data acquired from the experiment and configured with the same design parameters as the deployed system. The differences between ToF estimations were so small that it was impossible to visualize them, and most of the data points seemed to be overlapped. Then, Figure 14 presents the error among the ToF estimations shown in Figure 13. In the worst case, we had a maximal error of around $4.3\times 10^{-3}$ %, and the error increased with exposure time. Figure 15 shows how the signal level (SL) of the US response decreased during the experiment, explaining why the error increased. The SL refers to the root mean square (RMS) value of the first detected echo, measured after the amplifier [35]. This is an indication of the SNR of the digitized signal after bandpass filtering.

Next, we took the slope of the linear fittings shown in Figure 12 as an estimation of the corrosion rate at each measurement point. Afterwards, a filtered corrosion rate was obtained by running the average of the last four slopes as is shown in Figure 16. Thus, the intraday corrosion rate could be estimated from the ToF measurements taken every 6 h and had an approximate delay of 4 days.

As we can see from the thickness measurements, Figure 12 shows increments of thickness followed by a rapid and greater decrease in thickness. A similar behavior was observed even in a laboratory experiment with permanently installed ultrasound transducer: an increase in thickness followed by rapid thickness loss [26,44,45]. This was a consequence of the ToF measuring method that relies on the speed of sound being constant at all points of the steel sample. However, corrosion provokes a gradual degradation of a small layer of steel that reduces the speed of sound and acoustic impedance in that layer. At the beginning of the corrosion process, acoustic impedance is still large enough to allow for the ultrasound signal to bounce off the initial wall, while slower speed of sound produces an apparent increase in thickness. When the acoustic impedance decreases so much that the ultrasound signal bounces off the new wall, the result is a rapid reduction in total thickness. The net result is, of course, thickness loss (see Figure 12). Therefore, although the average corrosion rate calculated below is correct (considering positive and negative slopes), the positive instantaneous corrosion rates were overestimated using the ToF method.

The proposed experiment allowed for us to measure an average corrosion rate equivalent to 0.134 mm/year, which is in very good agreement with typically reported corrosion rates for uniform corrosion (0.1–0.2 mm/year). It is also possible to reduce the sample interval from the 6 h of the current experiment. The experimental setup could be modified to drastically reduce its power consumption by removing the Beaglebone board, resulting in a more frequent waking up of the sensor node using the same batteries. This means that ToF measurements can be taken, for example, every hour instead of 6 h, reducing the time response of the corrosion rate estimation algorithm.

## Figures and Tables

**Figure 1 sensors-22-08451-f001:**
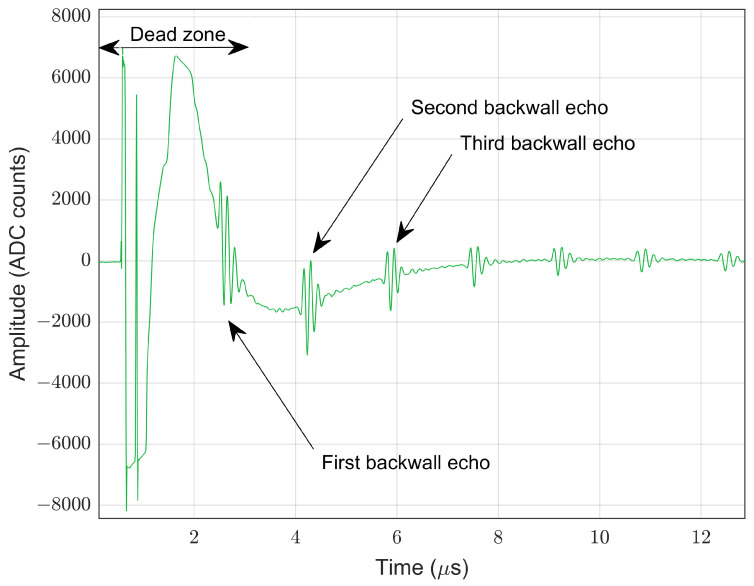
Raw piezoelectric signal with echo representation.

**Figure 2 sensors-22-08451-f002:**
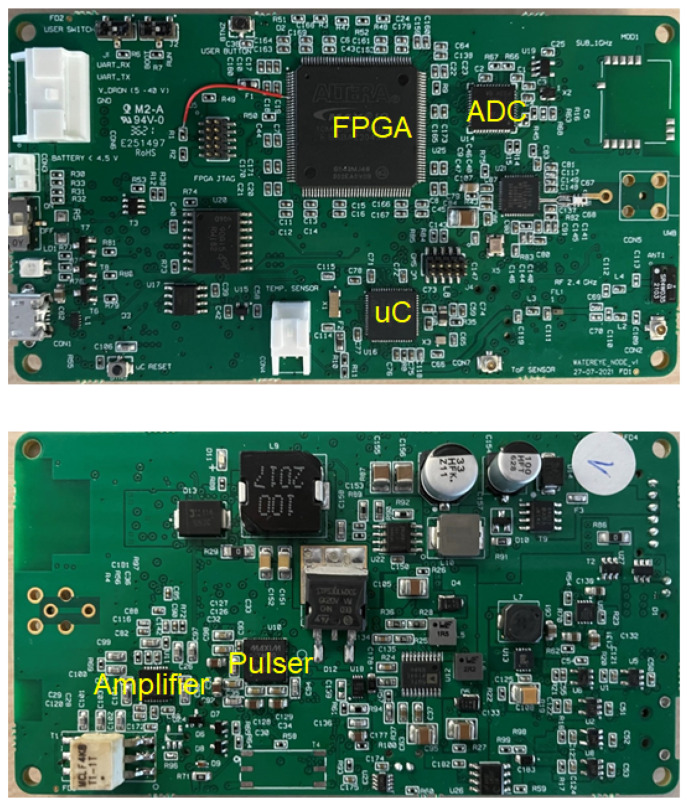
Electronics of the ultrasound corrosion sensor node.

**Figure 3 sensors-22-08451-f003:**
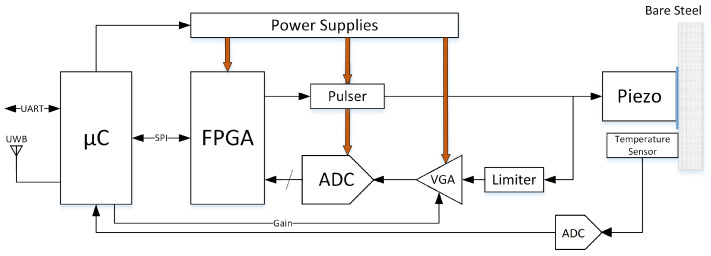
Hardware architecture of the ultrasound corrosion sensor node.

**Figure 4 sensors-22-08451-f004:**
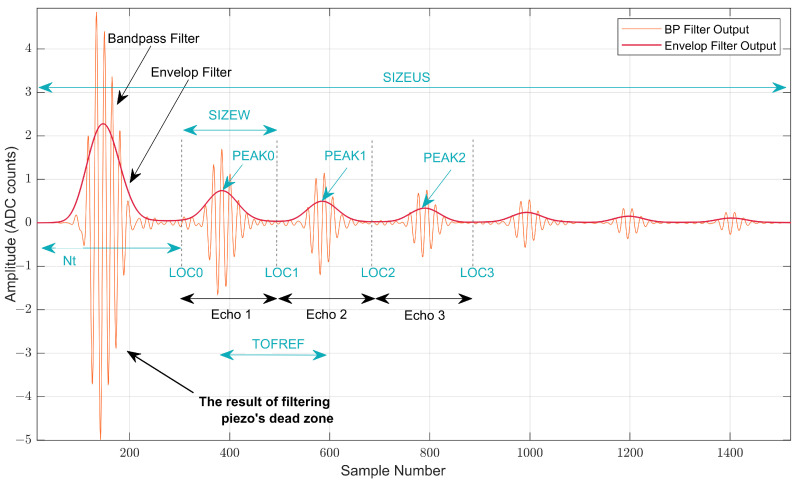
Design parameters used for output response signal analysis.

**Figure 5 sensors-22-08451-f005:**
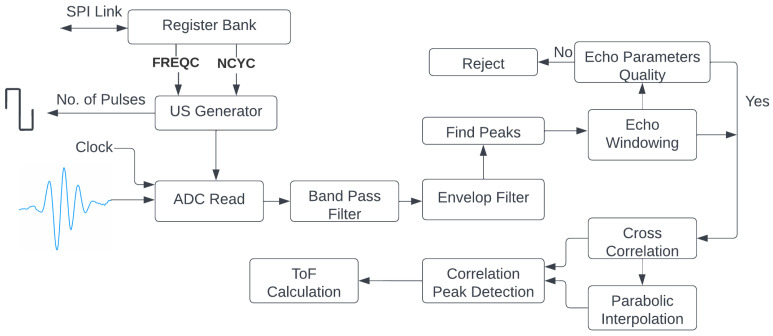
Block diagram of digital signal processing in FPGA.

**Figure 6 sensors-22-08451-f006:**
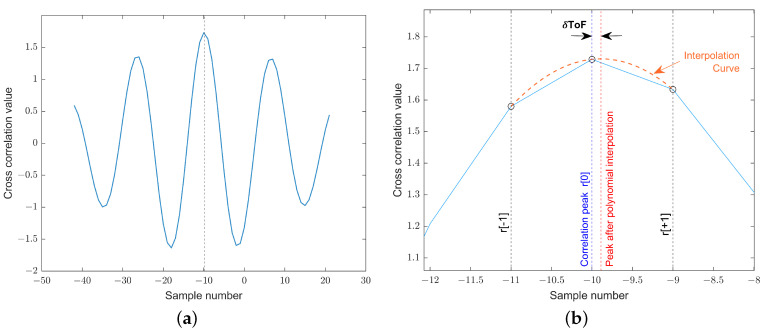
(**a**) Final cross-correlation output; (**b**) parabolic interpolation around the maximum of cross correlation.

**Figure 7 sensors-22-08451-f007:**
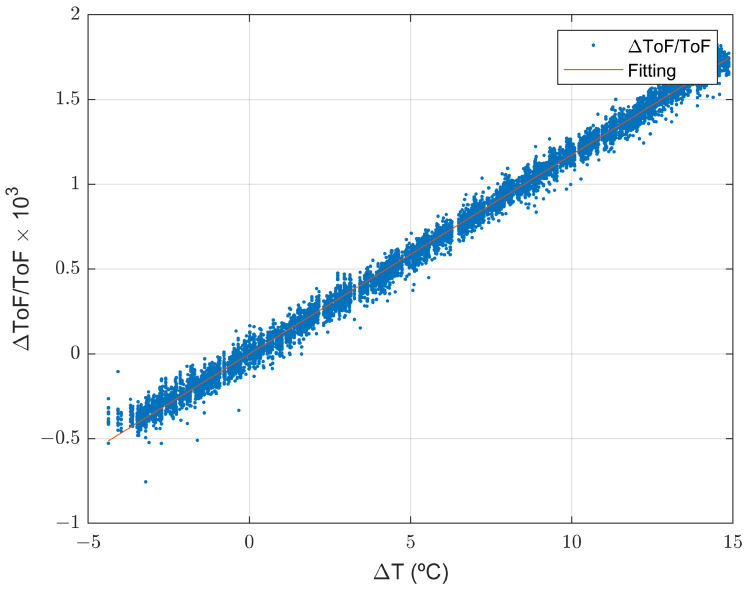
Experimental results to estimate the value of parameter ξ.

**Figure 8 sensors-22-08451-f008:**
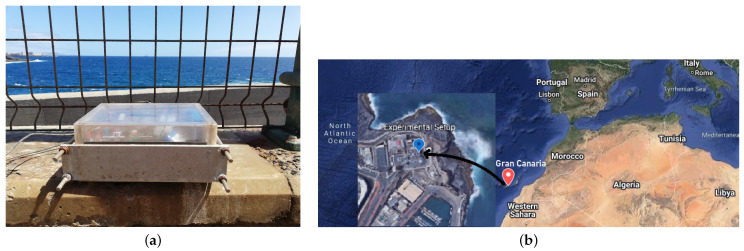
Location of the experimental setup. (**a**) Photo of the final location of the system (provided by PLOCAN); (**b**) experimental setup location on the map.

**Figure 9 sensors-22-08451-f009:**
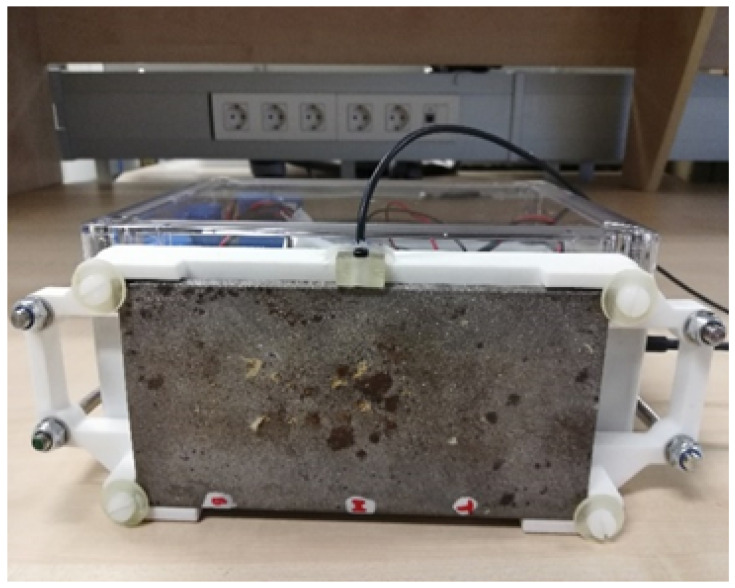
Bare steel sample attached to the monitoring solution.

**Figure 10 sensors-22-08451-f010:**
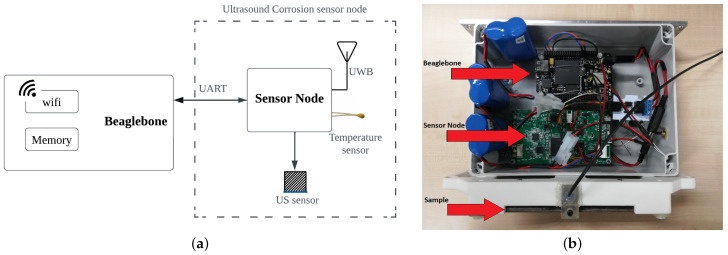
Final electronics setup used in the experiment. (**a**) Block digram of the setup; (**b**) photograph of the actual setup.

**Figure 11 sensors-22-08451-f011:**
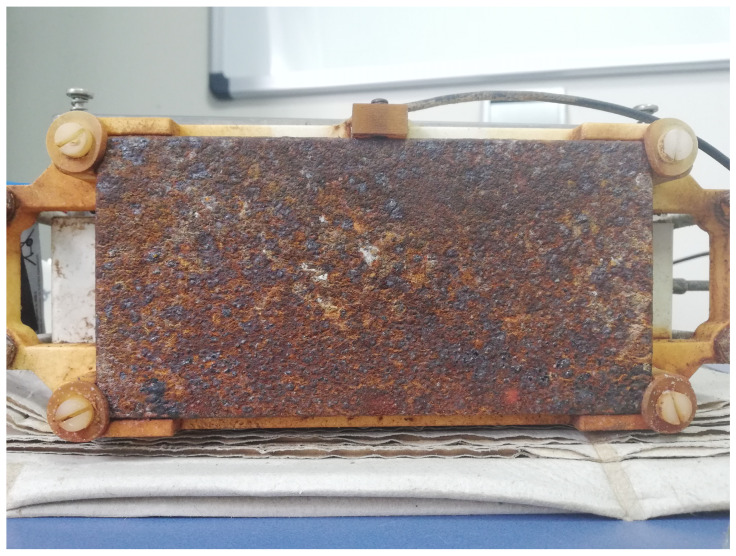
Photo of the bare steel sample after five months of exposure in the Gran Canaria (provided by PLOCAN).

**Figure 12 sensors-22-08451-f012:**
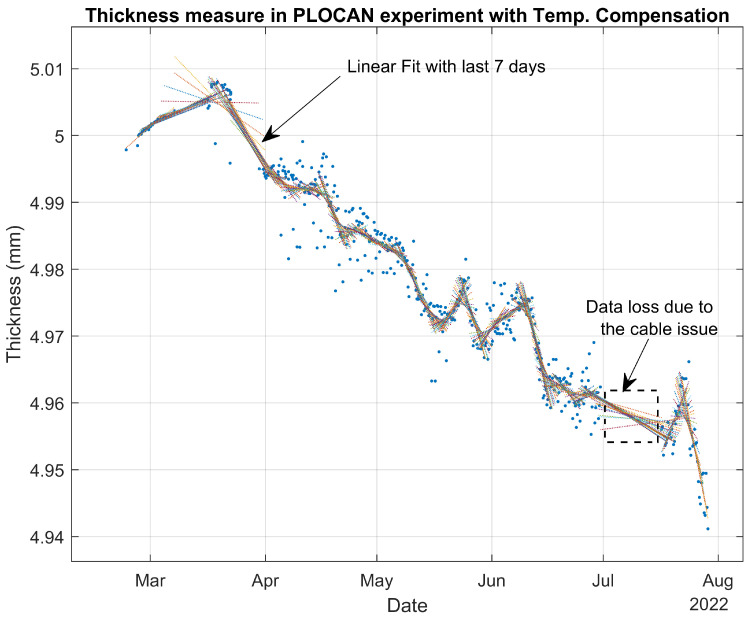
Thickness estimations from the experiment’s data with temperature compensation.

**Figure 13 sensors-22-08451-f013:**
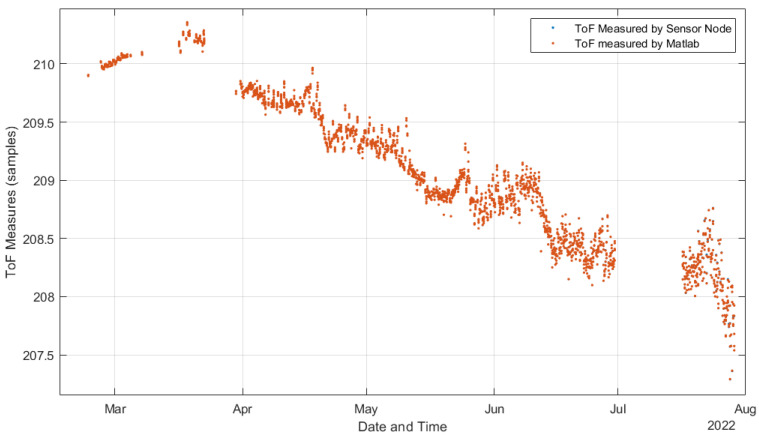
Comparison of ToF estimations with the MATLAB model.

**Figure 14 sensors-22-08451-f014:**
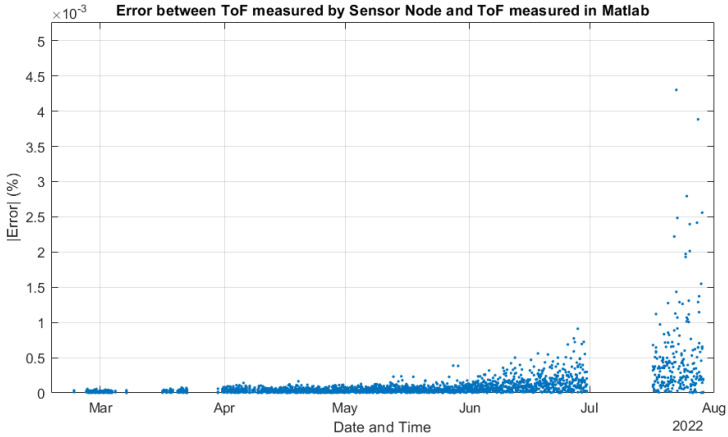
Error between ToFs estimated by the sensor node and by the MATLAB model.

**Figure 15 sensors-22-08451-f015:**
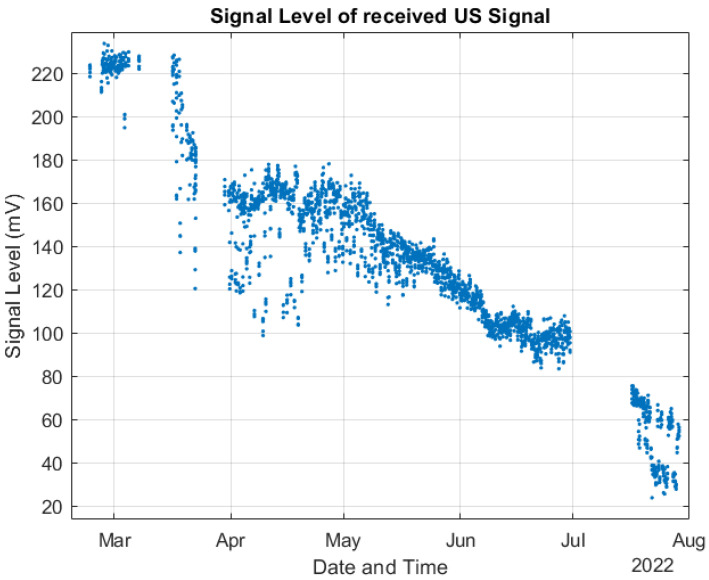
Signal level of the received US signal during the experiment.

**Figure 16 sensors-22-08451-f016:**
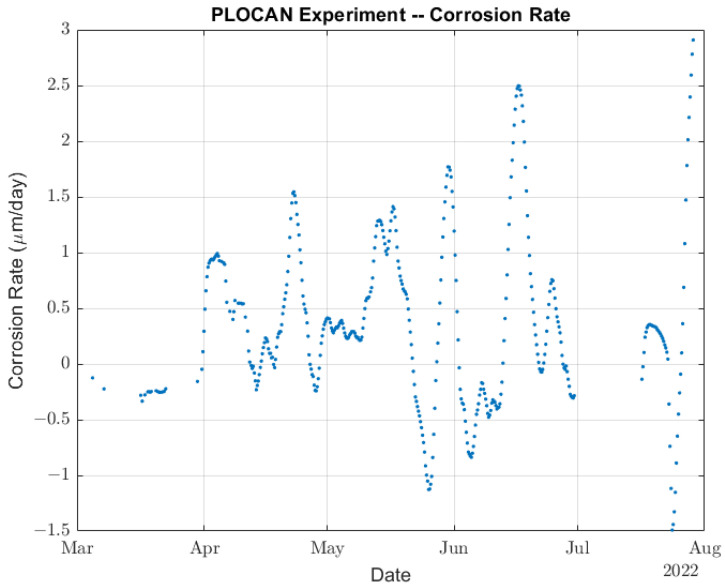
Estimation of the intraday corrosion rate.

**Table 1 sensors-22-08451-t001:** Parameters to configure the digital processing from the microcontroller.

Parameter Name	Description	Value
SIZEUS	Number of data samples received from the ADC	2048
NT	Number of samples that must be trimmed	0
	from the envelop signal to remove	
	of the piezo’s dead zone	
FREQC	Number of cycles of the sampling period	16
	to obtain the piezo sensor’s resonance	
	frequency	
NCYC	Number of periods of the excitation pulse	1
TOFREF	Approximate expected value of the ToF in number of samples	210
SIZEW	Typical echo duration in number of samples	192
ECHON	First echo to be detected and processed	2
SIZEWL	Number of cross-correlation samples taken on the	32
	left-hand side of the correlation index 0	
SIZEXC	Total number of processed cross-correlation samples	64

**Table 2 sensors-22-08451-t002:** Main features of the experiment.

Experimental Setup	Value
Sample material	S355
Sample thickness	5 mm
Ultrasound probe	V111 (Olympus)
Probe diameter of contact	15 mm
Probe nominal element size	13 mm
Adhesive/Couplant	Structalit 1028 R
US signal frequency	7.8 MHz
US signal amplitude	30 V (±15 V)
Speed of sound in S355 (long. waves)	5950 m/s
Speed of sound thermal coefficient ξ	$-1.054\times 10^{-4}\;$°C${}^{-1}$
Thermal expansion coefficient $c_{x}$	$12\times 10^{-6}\;$°C${}^{-1}$
Experimental location in latitude–longitude (decimal degrees)	27.9920, −15.3686
No. of measurement events per day	4 (one every 6 h)

## Data Availability

Not applicable.

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
