# Peer review of "Small Ultrasound-Based Corrosion Sensor for Intraday Corrosion Rate Estimation"

_sensors, 2022, doi:10.3390/s22218451_

Round 1

Reviewer 1 Report

The content of this paper was to introduce a novel theory and new design to detect the corrosion rate of metal materials in the marine environment. By the in-situ test, the performance of this device indicated the available function of corrosion detection. Some enhance the quality of this paper before it could be accepted as a research paper.

1.In Figures 1 and 9, Please provide the diagram of devices to replace the actual IC boards. It will present clear pictures of the novel design.

2.In Figure 13. Comparing the ToF estimations with the Matlab model, it is not easy to observe the difference between the estimation of the experiment (Xi) and the estimation done by the toolbox (Yi). Let us define the error as Ei = Xi – Yi. Please present the figures of Yi vs Xi and Ei vs Xi. Some significant errors will be found in these figures and please commend them.

3.In lines 321-323. “Once the experiment started we could see that the measured temperature was higher than expected due to the setup being exposed to direct sunlight.” That was a serious mistake in temperature measurement. Please describe the method to revise this mistake in the experiment in detail.

4.Figure 14. Thickness estimations from the experiment’s data with temperature compensation. It was not easy to understand the temperature compensation technique. Please describe the compensation technique in detail.

Reviewer 2 Report

The manuscript deals with interesting topic of remote sensing of metal corrosion. This topic is getting more and more relevant with the introduction of Structural health monitoring.  The presented approach with ultrasound based corrosion sensor is not so common and is definitely of interest.

There is however a serious shortcoming in this manuscript: the authors devoted a lot of attention to the description of the electronics, signal processing and data transmission. On the other hand, the author do not seem to recognize that the corrosion rate is never fully uniform over the area and corroded metal has surface roughness (different thickness at different locations) and the pristine metal is also covered with layer of rust (iron oxides, iron hydroxides) that have different composition and thickness. This is not even mentioned in the manuscript let alone being investigated what is the impact of various roughness and rust coverage on the detected thickness.

The manuscript should be shortened (made more concise, keep only the truly relevant content) in the description of the electronics, signal processing and data transmission. I understand that authors are very skilled in these fields.  What I would really like to see in this manuscript is an investigation how different thicknesses (roughness) of steel and different oxide layer thickness and composition underneath the sensor impact the response signal and what could be deduced from such signal and what is the uncertainty.  I understand that in the current case the rust layer is relatively thin compared to the 5 mm, however it should be taken into the account that rust layer could be easily several mm thick.  Does such rust layer still reflects the waves as at steel/air interface?  

Comments:

line 28: main roots à main routes

line 38: be led à lead to

line 46: in north sea ---  do you mean North sea ?    northern seas ? ??

line 68: “once-off” is not well defined, use some other term

line 69: “While the monitoring….”  --- this sentence does not seem to be finished, correct!

line 72: signals allows  à signals allowing

line 112: electric impedance à electrical impedance

line 126: accelarated à accelerated

section 2.2 – here you should tell what is the size (diameter) of the sensor. This explains what surface area you are actually measuring.

eq. (2) – is it necessary for understanding?  Or can you simply say that excitation pulse frequency is 7.8125 MHz ?

Table 1: are all these parameters relevant for understanding the sensor operation?  FREQC is not the freq. of the excitation pulse but Frequency Division Factor. Also, what are the units for ToFREF and other time-related values ?

Eq. (3)  -- is this equation really needed to be written ?

In section 2.2 there are too many details that do not contribute to understanding, many paragraphs could be shortened. However, I am missing what is the sampling frequency for the response signal?

For figure 3 it should be written that this is the response signal.

I appreciate that you described the piezo’s “dead zone”

Figure 4 should be presented before Figure 3, it would be much easier to understand.

Is the description of parabolic interpolation really so relevant?  Is the precision improvement obtained by parabolic interpolation essential for proper determination of corrosion rate?  How much is difference of one sample number in terms of ToF for 5 mm thick steel?

Temperature compensation should be described in a more concise way. Skip the eq. on top of page 10 (why certain equations do not have the number?).

Introduction and derivation of relative thickness loss could be easily omitted.

Section 3:

it is not written where the experiment has been carried out (submerged in the sea water, in the splash zone, on the land close to the sea….). It took me quite some time to realize that it is actually atmospheric corrosion (coastal region) that you monitored.

Did you use any couplant material?  If not, why not?

Line 277 and few further lines:  This seem to be the spot where authors attempted to justify thickness variation across the steel plate. However, for 13 mm diam (0.5 inch) sensor V111 it is highly unlikely that thickness would be uniform underneath the whole sensor (13mm diameter area).   Honestly, those few lines do not explain how the current monitoring approach deals with non-uniform thickness due to corrosion (+ different composition and thickness of rust layer attached to the steel surface).

line 294: "in a day, system takes five successive ToF measurement every six hours" – this is not possible in 24 hours, later on you state 4 measurements every 6 hours per day.

line 296:  as described before, state the location (under sea, splah zone, coastal region (atmospheric corrosion)….

Figure 11: it is obvious that you do not have uniform thickness and that the surface is covered with oxide (several types as can be deduced from color). 

Table 2: 2x probe diameter written (it should be given much earlier in the text). Experiment location needs units

Figure 12 and the corresponding text can be omitted.

Line 313: “data processed by the deployed system with the estimations obtained from processing in our Matlab model (our own toolbox) configured in the same way, that is, using the same design parameters” – what is this Matlab model (your toolbox) ?  Do you simulate data somehow?  Please explain this!   What is the difference between red and blue points in fig. 13?  Why are red dots (estimation done by toolbox) more scattered at the end of measurement period? Figure 13 needs a legend.

Could the increasing scatter of measured datapoints with time be due to the roughness and oxide layer thickness that also increase with the exposure time?

line 335: provoques – not an English word

Line 330---342: please explain your explanation of thickness increase and general scatter more in detail.  Do you have any measurement supporting your theory on reduction of speed and acoustic impedance in the “small layer” ? (I guess you meant oxide layer.

Round 2

Reviewer 1 Report

The content of the revised paper has been improved significantly.